# The Impact of the Social Determinants of Human Health on Companion Animal Welfare

**DOI:** 10.3390/ani13061113

**Published:** 2023-03-21

**Authors:** Sonya McDowall, Susan J. Hazel, Catherine Chittleborough, Anne Hamilton-Bruce, Rwth Stuckey, Tiffani J. Howell

**Affiliations:** 1School of Psychology and Public Health, La Trobe University, Bundoora, VIC 3082, Australia; 2School of Animal and Veterinary Science, Roseworthy Campus, The University of Adelaide, Roseworthy, SA 5371, Australia; 3School of Public Health, Robinson Research Institute, The University of Adelaide, Adelaide, SA 5005, Australia; 4Adelaide Medical School, The University of Adelaide, Adelaide, SA 5005, Australia; 5School of Psychology and Public Health, La Trobe University, Bendigo, VIC 3552, Australia

**Keywords:** pet, human–animal bond, five domains model, healthcare, socioeconomic

## Abstract

**Simple Summary:**

The role of the social determinants of health (i.e., physical, social and economic factors affecting human health) and their impact on companion animal welfare have not been fully explored. Through a social determinants lens, it is possible to improve the understanding of companion animal guardian challenges in managing their companion animal’s welfare needs. Considering the five domains of animal welfare in conjunction with the social determinants enables us to explore the impact of the social determinants of human health on animal welfare. This highlights the importance of multidisciplinary collaboration to achieve positive health outcomes for guardians and positive welfare outcomes for their companion animals.

**Abstract:**

The social determinants of health (SDH) focus on the social, physical and economic factors that impact human health. Studies have revealed that animal guardians face a range of challenges in attaining positive welfare outcomes for their companion animals, which can be influenced by socioeconomic and environmental factors. Despite this, there is a lack of research specifically exploring the relationship between SDH and animal welfare outcomes. Given that the SDH impact on humans, which in turn directly impacts on their companion animal, it is important to adapt an SDH framework for companion animal welfare by characterising the impact of the SDH on companion animal guardians in their attempts to care for their animals and, by extension, the associated welfare outcomes. This paper explores how these human health determinants may impact animal welfare and the possible challenges that may arise for the guardian when attempting to meet their companion animal’s welfare needs. By integrating the SDH with other key frameworks, including the five domains model of animal welfare, through multidisciplinary collaboration, this framework can be used to inform future programs aiming to improve animal welfare.

## 1. Introduction

Over the past 20 years, there has been increasing recognition of the integrative relationships between human health and animal welfare across both human and animal disciplines [1,2]. For example, one study [3] notes that improved human welfare can also be linked to improvements in animal welfare. In Western societies, the family unit structure has extended to include how we interact with companion animals [2]. Studies have, therefore, demonstrated the positive relationship between companion animal ownership and human health outcomes, such as reductions in anxiety, distress, depression, loneliness, disease prevalence and increased physical activity [4,5,6,7,8,9,10]. However, not all studies support these findings, with some indicating either a negative correlation or none at all between companion animal ownership and human health outcomes [11,12,13]. Furthermore, selection biases and the use of convenience samples that may not be representative of the general population should be considered when interpreting the results presented in the relevant literature [14]. Whilst there is a lot of research in the fields of companion animal welfare and human health, there is a limited amount of literature that explores the roles and impacts each discipline has on the other.

Two existing frameworks, One Health and One Welfare, aim to highlight the interconnections between humans, animals and the environment [15,16,17]. One Health focuses on the integration of these sectors for human health outcomes [16,18,19]. One Welfare is an extension of the One Health framework, incorporating animal welfare by emphasising positive interactions between humans and animals, as well as supportive animal management techniques, to enhance both animal welfare and human well-being [15]. Both One Health and One Welfare highlight the importance of integrated, multidisciplinary collaboration to achieve better human health and animal welfare outcomes along with the importance of the development of a strong global One Health workforce both in the animal and human health sectors [20]. These frameworks provide a key starting point to illustrate the importance of interrelationships across both human and animal fields, but neither identifies the underlying causes of poor human health and animal welfare outcomes, with the human factors being those identified within the social determinants of health (SDH) [15,20].

The World Health Organization (WHO) identifies the role of SDH as a concept to ‘*tackle the social, physical and economic conditions in society that impact upon health*’ [21]. The SDH include aspects of a person’s life, such as income, education, social support and employment [22], and other circumstances in which a person is born, grows up, lives, works and ages [23,24]. Research has identified that consideration of SDH in the planning and designing of health campaigns or programs is associated with decreased morbidity and mortality, reduced health disparities and improved population health in marginalised groups [25]. Given that humans share their social, political and physical environments with companion animals, it is reasonable to conclude that because these environments affect humans, they would also directly affect animals [25,26]. Historically, regardless of this close human–animal interaction, the human and animal sectors have traditionally worked in silos, thus not truly integrating public policy or service delivery to achieve positive outcomes for both humans and animals [27]. To achieve positive human health and animal welfare outcomes, it is necessary to consider a prevention-orientated approach encompassing cultural, economic and political factors that underlie the physical and social environments in human and animal populations [28], reflecting their shared experience [27]. Despite this, applying the SDH to companion animals to improve animal welfare outcomes has not previously been explored in detail.

Historically, the concept of safeguarding animal welfare has been defined as being achieved through the ‘five freedoms’, developed in 1979, with the later development of the five domains, in 1994 [19,29]. The 2020 model identifies the five domains as (1) nutrition, (2) physical environment, (3) health, (4) behavioural interactions and (5) mental state [30]. In earlier models, the five domains focused mainly on identifying and correcting negative welfare states, while the current model emphasises the presence of positive welfare states [29,30]. Whilst the five domains model remains an important part of evaluating animal welfare, it does not focus on factors related to the animal guardian that could lead to poor animal welfare.

The aim of the five domains is to enable a systematic and structured welfare assessment of the animal, taking into account both the negative and positive experiences that an animal may experience, commonly referred to as their affective states [31,32], along with how humans may impact these states [29]. However, the five domains and the five freedoms models (the antecedent of the five domains) [33] fail to explicitly include links to the human impacts related to the income, education, employment or social status of the animal guardian on animal welfare outcomes [32].

The SDH influence how humans engage with and care for companion animals, which can impact on animals’ welfare outcomes [26]. The most disadvantaged sectors of the human population often have poor health outcomes and lower life expectancy, and they face a range of socioeconomic challenges [22,24]. For instance, limited access to foods to meet daily nutritional needs can result in chronic health conditions, such as diabetes and heart disease [34]. For the community as a whole, safety and access to affordable goods and services play a role in health outcomes [34]. For example, both individual human and community determinants impact on the animal guardian’s ability to access public space to exercise the animal and feel safe enough in the community to do so, an issue which is especially relevant for dog guardians. Companion animal guardians who come from lower social–economic communities may face many obstacles, such as the cost of veterinary care, cultural and language barriers, veterinary–client communication challenges, lack of accessibility of care and lack of education, all potentially influencing the animal guardian’s ability to provide positive animal welfare outcomes [19,35,36,37].

We integrated the SDH with the other frameworks through a review of the existing literature and the consolidation of the concepts pertaining to the interrelations of companion animals and their guardians. By adapting an SDH model to address the impact of humans on animal welfare outcomes, based on the existing frameworks for human health, this will enable us to identify how the social determinants of health in human guardians may influence animal welfare outcomes. This will enable the development of preventative policies to protect the human–animal relationship and enhance positive welfare outcomes for companion animals. The aim of this paper was to characterise the impacts of the SDH on companion animal guardians in their attempts to look after their animals and, by extension, the associated preventative measures to mitigate potential negative animal welfare outcomes. Note that in this paper the term ‘guardian’ rather than ‘owner’ is used. Whilst we respect that the term ‘guardian’ in human literature often refers to a temporary carer, the term ‘guardian’ or ‘guardianship’ is often used to reflect the relationship between humans and animals [38]. Rather than the animal being ‘owned’ by a human as reported within most legal frameworks, the authors choose to lead by example, as they consider that animals should not be property and instead support the view that humans are guardians of the animal’s welfare [38].

## 2. Impact of the Social Determinants of Health on Companion Animal Welfare

Regardless of socioeconomic status or background, a deep connection can develop between guardians and their companion animals [39]. However, this connection can also bring challenges for the guardian in ensuring the well-being of their companion animals [39]. Studies on the role of policies related to animal populations and their effect on human inhabitants and government organisations identified that animal shelter intake per capita was higher in neighbourhoods with lower mean household incomes, education and housing stability. The number of businesses that support companion animal ownership, such as grooming, veterinary care, behaviour support and kennels, was also lower in neighbourhoods with a higher animal shelter intake. Studies also identified that neighbourhoods with higher violent crime rates had a higher reported incidence of animal abuse, dangerous animals and illegal dog fighting [40,41,42,43,44]. Dogs involved in organised fights suffer from physical pain and also psychological and emotional distress, indicating that the actions of the guardian can influence the welfare outcomes of the dog [41]. The quality of infrastructure in a region can have a significant impact on both companion animal and guardian health by influencing the quality of veterinary care. It is highly plausible that if the SDH for humans were improved, so too would animal welfare [45].

The challenges of healthcare costs were demonstrated during the COVID-19 pandemic, where socioeconomic factors contributed to humans delaying accessing their own and their animals’ healthcare, especially in scenarios where the human had low levels of social support or was not able to afford animal care services [46]. The same was found in the wake of Hurricane Katrina, when companion animal owners refused treatment, shelter and healthcare for themselves to ensure they were not separated from their animal [19]. This emphasises the influence that companion animals have on human healthcare decision making and the importance of an interagency response [46]. Unfortunately, the relationship between humans and companion animals is often neglected when services are responding to society’s basic needs [47]. For example, if a person does not have existing support and is homeless, it has been reported that they will forgo accessing medical care due to the lack of someone who can look after their companion animal [47]. Whilst this highlights the importance of the role of companion animals in their relationship with human guardians and subsequent health outcomes, research investigating the role of SDH on companion animal welfare is limited.

Studies focusing on the link between human social vulnerability on dog intake to an open shelter found that when humans and dogs share the same social and physical environments, the same vulnerabilities can be found across both population groups [27,35,48]. As such, companion animals that show signs of social and physical neglect are likely to be an extension of the same challenges faced by their guardian [27]. In particular, communities that experience socioeconomic challenges and are socially vulnerable are at a high risk of companion animal abandonment [27]. To enable companion animal guardians to achieve positive welfare outcomes, both the guardian and the animal risk factors need to be understood and integrated as one key focus area to develop clear strategies to prevent relinquishment [49,50,51,52,53], an approach supported by utilising the SDH in relation to companion animal welfare. Therefore, adopting an integrated approach to both the guardian and the animal in the context of the SDH can facilitate positive welfare outcomes for companion animals.

We used the United States Department of Health and Human Services SDH domains model to explore how the SDH influence animal welfare through the ability of companion animal guardians to provide care for them [54]. The five key SDH domains in this model are education, healthcare, environment, social/community and economic stability [54]. Each of the five SDH domains were explored to identify the relationship between humans and companion animals, as well as the potential negative animal welfare outcomes, as demonstrated in Figure 1. For the purpose of this review, the focus of Figure 1 is on the potential negative outcomes of SDH rather than the potential positive outcomes. There is also substantial overlap across the domains, and they do not exist in isolation; rather, all domains should be considered simultaneously as a holistic phenomenon for guardians and their companion animals.

## 3. Applying the Social Determinants of Health to Companion Animal Welfare

In this section, we describe each SDH domain in turn, explaining how it can indirectly impact companion animal welfare. These domains were treated individually for the purposes of this review and for the sake of clarity. However, they are strongly interconnected and should be viewed holistically when considering an individual guardian’s ability to care for their companion animal (e.g., a person’s economic stability will inform their ability to access a safe environment to live in, and the ability to access high quality education will also inform their economic stability). Although we tried to avoid overlap as much as possible in this review, it was impossible to prevent it completely due to the fact of this inherent interconnectedness.

### 3.1. Education

The SDH domain of education is linked to factors such as literacy, language, vocational and higher education (Figure 1) in humans. Within the companion animal context, this underpins the guardian’s ability to understand how to train an animal and the importance of preventative health such as vaccinations (Figure 1). A study into the reasons for relinquishment at various animal shelters within the United States identified that guardians who had not reached an educational level beyond high school were more likely to surrender an animal [55]. Communities with a higher educational attainment had lower stray intake at animal shelters [56]. Guardians with lower levels of education were less likely to have visited a veterinarian within the last 18 months (57.1% with a high school education, 80.3% with a bachelor’s degree) [57]. However, the companion animal guardian’s ability to access services and/or support programs appears to be influenced by other associated factors, such as transport barriers, availability and hours of operations of veterinary/training services and affordability [58], highlighting the interconnected nature of all of the SDH domains.

Not understanding behavioural challenges in companion animals is a common reason for animal abandonment, relinquishment and euthanasia [55,58,59,60,61,62]. Studies have reported that between 22% and 35% of dogs are relinquished due to the fact of behavioural issues/concerns [50,63]. One preventative aspect of negative behavioural problems that could possibly avoid relinquishment is the early socialisation and training of companion animals, which is most commonly achieved through puppy classes [64,65,66]. A twenty-week study of participants who attended early animal socialisation classes compared to those who did not identified that 99% of the companion animal guardians who engaged in early socialisation methods had achieved a high school level or above qualification, suggesting a link between education and understanding the need for early animal socialisation [64]. In a study of rehomed pets, 34% of the participants identified that free or low-cost behaviour training would have prevented the relinquishment [67], with 70.6% reporting the same in a more recent study exploring the reasons for the relinquishment of dogs at a shelter [67]. Guardians are also unlikely to seek help for behavioural issues, with one study identifying that just over 50% of participants were not very likely or not at all likely to access veterinary care for these issues [57]. However, another study identified that access to low-cost/free behaviour consultation (67%) would assist in preventing relinquishment [63]. Barriers to attending such training or companion animal classes include the guardian’s lack of motivation to attend, limited awareness, expense, class size and geographic availability [68].

### 3.2. Healthcare

The SDH domain of healthcare is linked to the availability and coverage of health-related services, cultural and linguistic competency, and quality and affordability of care (Figure 1). Understanding the impact that human health conditions have on animal welfare has received limited attention. A few studies have identified companion animal guardian health reasons for relinquishment [61,69,70,71]; in particular, a Danish study identified that health-related conditions of the companion animal guardian was the primary reason for relinquishment at a large Danish shelter [72]. This was also reported in an earlier study [70,73], but neither study explored what the health-related illness/conditions of the companion animal guardian actually were or if they were related to any particular demographic age groups [60]. Allergies associated with having pets have been raised in some studies, with the majority of these studies linking allergies to cat guardianship [62]. Whilst the understanding of the exact types of healthcare conditions beyond allergies is scarce, some parallels can be explored through studies with assistance dogs. A handler’s medical condition has been found to potentially have an effect on the companion animal’s welfare, both in the short and long term [74,75], resulting in the return of the assistance dog to an organisation or even relinquishment [76].

### 3.3. Environment

Given that humans and animals that live together share the same environment, the SDH environment domain highlights the challenges for low-socioeconomic communities in accessing services for their animal due to the fact of limited transportation options and safe access to green spaces to provide the animal with exercise and enrichment (Figure 1). The environment of companion animals and their guardians is variable depending on the region in which they live, along with traditional cultural and societal norms; therefore, SDH need to be considered in line with the region and culture in which it is being applied to ensure the outcomes for companion animals are proportionately reflected. Most of the literature in this section relates to the United States and Australia. Transportation is an important challenge for companion animal guardians in providing positive welfare outcomes for their animal [44,57]. In most cities in the United States [77] and throughout Australia [77,78,79], access to public transportation with a companion animal is typically limited to registered assistance animals or animals small enough to fit within a cage, although European countries often permit companion animals on public transport [77]. As a result, most companion animal guardians, especially those in low-socioeconomic areas in the United States and Australia, are unable to utilise public transportation with their pet [36,80,81]. The distance travelled also has an effect on companion animal guardian’s ability to attend early socialisation classes, with those who live in rural or remote areas being 2.5 times less likely to participate [64].

Companion animal guardians may also face challenges in enabling their companion animal to access parks to support exercise and positive social interactions with other people and their companion animals [27]. Using the Social Vulnerability Index (SVI), it was found that households located in areas with a high SVI had difficulty socialising their companion animals, as the companion animals were often left alone for extended periods of time [27]. Companion animal guardians can have a direct influence over exercise and associated animal welfare outcomes, but if the community in which the companion animal and guardian reside is unsafe, such exercise opportunities might be limited. A review of the usage of green spaces within disadvantaged communities identified that most studies across the literature cite safety in accessing community green spaces as a key concern underpinning their limited use [82]. It has been reported that in the United States, behaviours that make another feel unsafe in their environment within low-socioeconomic communities was twice that of higher socioeconomic communities [83]. Aside from safety considerations, green spaces in lower social–economic communities are equipped with fewer amenities [84,85], are located further away from individuals’ homes [86] and are less accessible than those in higher socioeconomic communities [87], thus restricting the ability of the companion animal guardian to provide positive welfare opportunities.

One of the many barriers to securing housing is having a companion animal and the reluctance of landlords or housing management organisations to allow applications from companion animal guardians [35,44,63,72,88,89,90]. For animal guardians trying to exit homelessness, pet ownership is the biggest barrier [81,91,92,93,94]. One study reported that 42.1% of participants relinquished their pet due to the fact of moving when the landlord would not allow companion animals [88]; these numbers were even higher in another study, with 77.5% relinquishing their pet due to the fact of moving and 35.1% due to the fact of landlord conditions [63]. It is reported that between 5% and 25% of people who are unable to secure housing in the United States are companion animal guardians [91,92,93,95]. This challenge goes beyond the stereotypical socioeconomic barriers and affects a large swathe of the population, who have to choose between a roof over their heads or the relationship with their companion animal [96]. This is particularly prevalent in the current challenging economic environment, whereby rental availability is at an all-time low in many countries, including the United Kingdom [97], United States [97] and various cities in Europe (e.g., Amsterdam, Lisbon and Athens) [98], further limiting the availability of options for housing for companion animal guardians [99,100]. Whilst some states in Australia, such as Victoria, Queensland and Australian Capital Territory, have passed legislation preventing tenancy agreements from banning companion animals, animal guardians in other Australian jurisdictions are denied this protection [100]. Other factors identified as preventing relinquishment include pet-friendly housing (33% of participants), temporary boarding animal care (30% of participants) and pet-related housing deposits (17% of participants) [67].

The ability to access veterinary care and veterinary-related services, such as grooming and behavioural training, is a substantial barrier for low-socioeconomic populations and has a direct effect on the health and welfare of companion animals [25]. Companion animal guardians’ access to veterinary care, including animal and human transportation, is further challenged by the availability of veterinary services. The availability of veterinary services within low-socioeconomic populations is limited. From a business perspective, a private practice is unlikely to set-up in an area where they are unable to make a profit or where companion animal guardians are either unable or unwilling to pay for veterinary care [36,101]. This finding is supported by a study in which for a low-socioeconomic area of Chicago (United States), which comprises 25% of the city’s population, only 7% of the city’s companion animal services were based in this area [40].

### 3.4. Social and Community

Where we live, work and learn, along with community involvement and equality, all play a role in improved health outcomes for humans [102]. Within the animal context, the animal guardian’s ability to provide a safe place and their relationship with the animal and community, along with support networks, all factor into the overall context of enabling positive welfare outcomes for the companion animal (Figure 1).

In communities where environmental factors enable social and community engagement (see Section 3.3), companion animal guardians, such as those who are able to walk their dogs, are more likely to get to know people within their community than those who do not have a companion animal [7]. Furthermore, approximately one-quarter of the respondents reported that they met people within their community as a result of their companion animal and are part of their social circle, with 42.3% having received social support from people they met through their companion animal [7].

While there is evidence that companion animal guardians living in these positive environments can develop meaningful friendships with others in their community, as noted above, it has been reported that lower socioeconomic communities have poorer social networks and social support [103]. This limited social support has been found to delay healthcare treatment in companion animal guardians in low-socioeconomic communities due to the fact of financial constraints that inhibit care for the animal [104]. In addition, access to services required for animal guardians also includes the ability to be culturally aware and/or provide translation services [25,105].

While a person’s social and community life with their companion animal extends beyond family, especially violent family situations, it has been well established throughout the literature that there is a link between domestic violence and animal abuse [47,106,107,108,109,110,111,112,113,114,115,116,117,118,119,120]. However, the challenge presented to most survivors of domestic violence is their ability to leave their home and take their companion animal(s) with them [47]. Studies have shown that between 26% and 71% of female companion animal guardians experiencing family violence reported that the offender had seriously harmed or killed the companion animal [107,109,113,114,118,121,122,123,124,125]. Many survivors of domestic violence (48%) are hesitant to escape their domestic violence environment due to the fact of being concerned about what will happen to their companion animal [126]. Further, it has been reported that approximately 18–48% of survivors have delayed entering a domestic violence shelter due to the presence of welfare concerns for their companion animals that they had to leave behind [47,109,113,121,127]. As a result, both companion animal and guardian safety and welfare is at risk, as most survivors remain within the domestic violence environment, not wishing to leave their companion animal behind and having limited safe refuge options that accept guardians with companion animals [106,107,128].

### 3.5. Economic Stability

Income is considered to be one of the most influential of the SDH [26] along with employment, debt and expenses (Figure 1). For companion animals, this domain focuses on the guardian’s ability to access the required resources to provide for the animal’s needs cognitively, physically and environmentally. This includes their ability to pay for associated veterinary costs to ensure positive welfare outcomes, along with spending adequate time with their animal. However, some people with low incomes may have the possibility of spending more time with their companion animal if they have fewer other demands on their time (e.g., those on disability or retirement pensions) (Figure 1).

The issue of affordability for veterinary care as a barrier to maintaining companion animal guardianship has been covered extensively [35,36,57,67,80,129,130,131,132,133,134,135,136,137]. The most common reason provided across the literature for the relinquishment of a companion animal is low income [35,51,63,88,138,139], although the reasons for relinquishment continue to be multifactorial. Within Australia, areas of greater socioeconomic disadvantage had a higher number of preventable Canine parvovirus cases and higher rates of euthanasia-without treatment for parvovirus, but the reasons for this disparity were not fully explored [140]. The lack of affordable vaccinations and the number and timing of vaccinations are suggested to be causes, along with the ability to access veterinary services, thus resulting in poorer welfare outcomes for companion animals [141,142,143]. In a study of rehomed companion animals, 40% of the participants identified that free or low-cost veterinary care could have prevented relinquishment [57]. Furthermore, in a New-York-based subsidised grooming program, more than half of the animal guardian participants reported that the cost associated with companion animal grooming was a barrier to maintaining the animal’s welfare [144]. There is limited financial support available for veterinary costs, with most financial support only available in emergency situations [47]. One study identified that when affordable veterinary care is provided to low-socioeconomic companion animal guardians, the number of veterinary visits increase for both disease/injury and wellness, including preventative interventions, such as heartworm and vaccinations [145]. Understanding and addressing the economic barriers to both the access and provision of care for companion animals are important for the improvement of health and welfare outcomes [25].

Traditionally, across the literature, companion animal healthcare has focused on the sterilisation of animals, with a limited scope on the comprehensive healthcare needs of the animal [130,132,146,147,148]. Nonetheless, the ability to access free or low-cost spay and neuter services was identified in two studies as a factor that may prevent animal relinquishment according to 30% [67] and 53% [63] of the participants. Low-cost standard veterinary care was also identified as a preventative measure for 56% of the participants [63].

Companion animal guardians with limited income often find it difficult to obtain appropriate food for their animals. The Foodbank Hunger Report 2022 [149] highlighted that over half a million people in Australia are struggling with the cost of food; of this population, 67% have pets. This provides a challenging dilemma for companion animal guardians forced to choose between feeding themselves or their companion animal [150,151]. In studies investigating the impact of the cost of living on animal relinquishment, between 30% [67] and 50% [63] of the participants reported that having low-cost or free pet food available would have prevented them from relinquishing their animal.

Unlike human healthcare, where free or heavily subsidised human healthcare is available in many developed countries [152], there is no policy supporting a universal healthcare system for veterinary care, so it is necessary for guardians to cover the full cost of care. Most companion animal guardians have little understanding of the costs associated with both human and veterinary care, so they tend to be predisposed to the view that the cost of veterinary care is too high [153]. Pet insurance has been viewed as expensive [154], and there are limited marketing and education programs around insurance products, resulting in their limited use [154]. Furthermore, for financially constrained companion animal guardians, the ability to pay for veterinary care is already a challenge in addition to the payment of an insurance premium [153]. There are some current initiatives towards mitigating this financial challenge. For instance, a political party in Victoria, Australia, has identified the need for affordable veterinary care in Australia and is campaigning for universal healthcare for animals [155]. A pilot program in the United States, Program for Pet Health Equity, is an integrated interagency approach with social workers, veterinary services and financial support organisations to enable the support of animal guardians with a low-socioeconomic status [156]. The provision of financial assistance through AlignCare for animal guardians in receipt of public assistance was established at the University of Tennessee, whereby clients are responsible for a 20% copay at the time of the visit, and the remaining 80% is funded by AlignCare [156]. In addition to the copay arrangement, AlignCare has partnerships with animal welfare agencies to support guardians where payment may not be possible [156].

## 4. Five Domains of Animal Welfare and Their Relationship with SDH

The five domains model was developed as a scientific approach to evaluate animal welfare and promote positive outcomes [30]. The SDH can form part of a multidisciplinary perspective in conjunction with the five domains of animal welfare (See Figure 1) to improve companion animal welfare. Consider, for example, one aspect of the five domains, behavioural interactions, which explores an animal’s interaction with the environment, with other animals and with humans [30,157]. The SDH conditions of the companion animal guardian may result in a restricted or confined environment or limited animal-to-animal activity, exposure of the animal to threats or result in a guardian who is inexperienced and unskilled in animal behaviours and training methods [158]. All five social determinants (i.e., education, healthcare, environment, social/community and economic stability) can impact a companion animal guardian’s ability to provide training for their companion animal, which would have an associated impact on the animal’s five domains—behavioural interactions outcomes (as outlined in Figure 1). This can be further explored on the basis that companion animal guardians from lower socioeconomic communities may not be able to train their animals or access appropriate veterinary services due to the fact of limited financial means or transport restrictions. Furthermore, for companion animal guardians residing in a household where violence or abuse is present, this may result in companion animals being exposed to stressors with potential threats and physical harm. The impact on the five domains—behavioural interactions is only one example; the influence of the SDH can be found across all five domains.

The five domains of animal welfare framework also highlights the importance of agency [30]. Agency plays a key role when considering the influence of the SDH, as it reflects an animal’s natural tendency to interact with its physical, biological and social surroundings beyond that of its immediate needs, allowing the animal to make conscious choices to behave in a particular way [159]. However, the human’s ability to make informed and meaningful choices and exercise controls over their own life decisions is referred to as autonomy [160]. However, unlike human autonomy, animal agency is reflected in the companions animal’s ability to respond to certain stimuli, which can be influenced by the animal’s guardian and their level of autonomy [159]. This results in the animal guardian having a significant role in the impact of the agency of the companion animal, which in turn can be influenced by impacts of the various factors within the SDH. The level of agency and autonomy that a companion animal has is intertwined within their social and environmental factors, along with the ability, decisions and actions of the companion animal guardian. For example, if a companion animal guardian is able to provide access to exercise, positive reinforcement training and animal-to-animal and human-to-animal socialisation, the companion animal will have more agency and independence, which supports them in building their confidence and comfort within their environment.

## 5. Illustration of Social Determinants of Health on Companion Animals

The scenario below (Figure 2) demonstrates that there are multiple interrelated factors of the SDH that potentially have an influence on companion animal welfare outcomes. The illustration reminds us that not every companion animal guardian has the same opportunities to achieve the perceived optimal animal welfare outcomes, but the challenges faced by companion animal guardians have an effect on the animal’s welfare. This provides an evidence base from which to explore the development of service provision and/or public policy to achieve positive companion animal welfare outcomes.

## 6. Future Directions for Research

The SDH are designed to be comprehensive, taking a holistic perspective of the factors influencing a person’s health status. In this review, we extended this concept one step further by describing how the SDH can indirectly influence the welfare of companion animals in the care of their guardians. Despite the comprehensive nature of the SDH framework, there are some aspects of companion animal care that were beyond the scope of this review, and they merit further investigation. For example, there is evidence that anthropomorphism by guardians may have negative effects on companion animals [162], and the caregiver burden on companion animal guardians can affect their perception of health, prognosis and euthanasia [163]. Furthermore, attention should be paid to the role of the veterinarian and the need for capacity building programs so that veterinarians themselves understand the impact of the SDH on their patients [20]. Future research should consider these additional influences on animal welfare.

## 7. Conclusions

The integration of the social determinants of health into human and animal welfare is crucial for multidisciplinary public policy and preventative support provision. The proposed integrated framework considers the interconnection between the social determinants of health, One Health, One Welfare and the five domains of animal welfare, providing a model for a more comprehensive approach to better outcomes for companion animals, their guardians and the community. Understanding the influence of these factors (i.e., income and access to veterinary care, education levels, community involvement and equity) on humans and, consequently, their companion animals enables the development of interventions aimed at enhancing the welfare outcomes of both the companion animal and their guardian. By utilising this model, we can better understand how to protect the human–animal bond, improve animal welfare outcomes and achieve long-term success in keeping companion animals and their guardians healthy and happy together.

## Figures and Tables

**Figure 1 animals-13-01113-f001:**
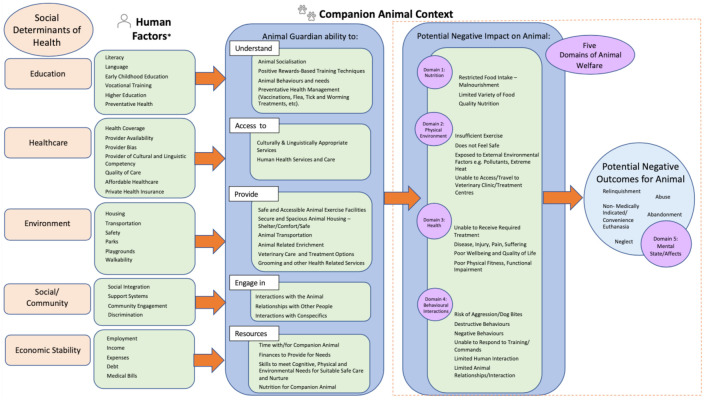
Five domains of SDH that influence animal guardians’ ability to care for their companion animal(s) and the potential negative outcomes on animal welfare. * Adapted from ‘Healthy People 2030′, US Department of Health and Human Services, Office of Disease Prevention and Health Promotion.

**Figure 2 animals-13-01113-f002:**
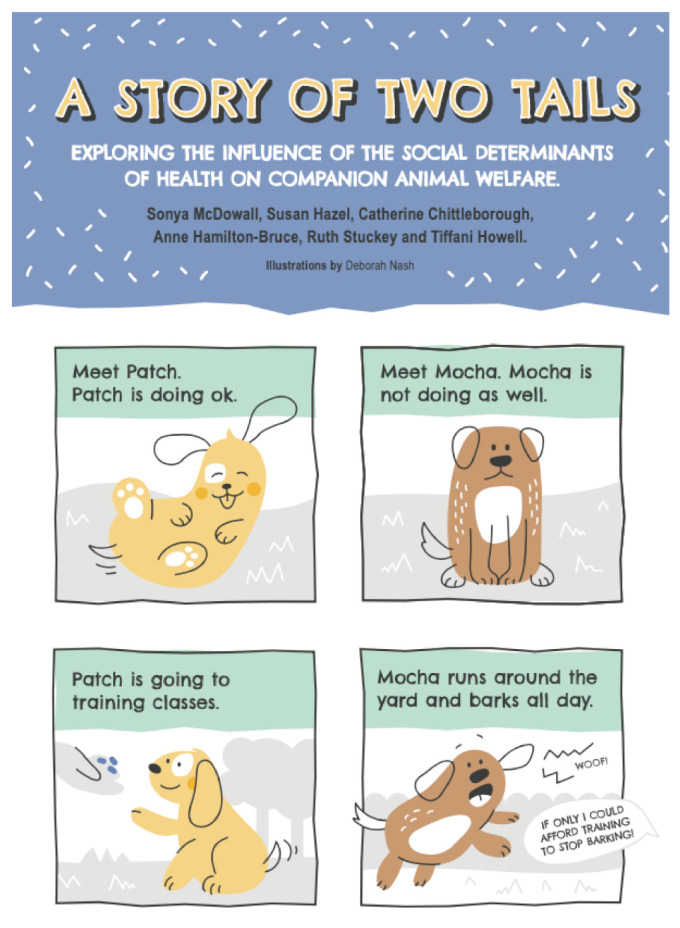
Story of Two Tails: exploring the influence of the social determinants of health on companion animals. Inspired by The Pencilsword: On a plate [161].

## Data Availability

Not applicable.

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
