# Peer review of "The Impact of the Social Determinants of Human Health on Companion Animal Welfare"

_animals, 2023, doi:10.3390/ani13061113_

Round 1

Reviewer 1 Report

The Impact of the Social Determinants of Human Health on Companion Animal Welfare

I want to commend the authors for such an interesting and relevant topic. The importance of assessing human factors that are directly or indirectly related to animal welfare outcomes was described in the manuscript, and since human factors comprise several issues, the complexity of the subject was adequately portrayed in this review. I left some comments hoping they can be useful for the authors.

Line 38: Consider replacing “pet” by “companion animal”.

Lines 59-61: The role of the veterinarian and the importance of capacitation programs regarding One Health One Welfare could also be highlighted (https://doi.org/10.1016/j.onehlt.2022.100382).

Line 61: Add a reference.

Lines 63-65: It would be interesting to briefly mention if, apart from human health, anthropomorphism could alter animal welfare outcomes (please, revise https://doi.org/10.3390/ani11113263

Lines 121 and 124: It might be better to just cite reference 37 at the end of the idea (at line 126).

Lines 124-126: Please review the following articles to add some additional information about the welfare of fighting dogs https://doi.org/10.3390/ani12172257 (The Welfare of Fighting Dogs: Wounds, Neurobiology of Pain, Legal Aspects and the Potential Role of the Veterinary Profession).

Line 126: I recommend adding if the infrastructure of the region can influence the quality of veterinary care, having an impact on both animal and human health.

Line 147: Describe which vulnerabilities are the authors referring to.

Line 154: Please complete this sentence by attaching information from the following articles: doi: 10.3390/ani5020364 (Epidemiology of Dog and Cat Abandonment in Spain (2008-2013) and http://www.doi.org/10.14202/vetworld.2021.2371-2379 (Abandonment of dogs in Latin America: Strategies and ideas).

Lines 174-177: The sentences mention that “Studies…”. Please, provide more than one reference.

Line 229: Although this section might refer solely to physical health, a mental issue such as caregiver burden has been shown to highly affect the way guardians perceive the health, disease outcome, and euthanasia decision of companion animals. Some articles that might be helpful are https://doi.org/10.1111/vde.13065 and https://doi.org/10.1177/1098612X221145835

Line 231: Since the aim of the present review does not mention a limit for the region (USA or Australia, for example), a paragraph could be added mentioning that this is one of the reasons why animal care can be limited in some underdeveloped countries. This is relevant because, therefore, the SDH model needs to consider the region to accurately assess the potential negative outcomes for companion animals (and this also has some social and cultural implications).

Lines 258-260: This information was already mentioned in lines 124-126.

Lines 436-442: Please, further explain what you mean by positive training and socialisation.

References. Amend according to the Instruction for Author’s Guide.

Reviewer 2 Report

Fascinating and long overdue integration of human and animal health frameworks. I have a few specific comments and suggestions below. However, I’m finding the headings and flow to be a bit disjointed and unclear. For example, 2. Applying the Social Determinants of Health to Companion Animal Welfare This section seems to include both what SDH is as well as how it might impact the animal. And also that social vulnerability extends from humans to their companion animals.  It then goes into the SDH components in detail.  I recommend a high level overview of SDH first as well as the 5 Domains—chunks of the section on what the five Domains are (the first paragraph in section 3 would fit well up here) should go here to set the stage for the two frameworks of interest in this manuscript (if these are the main ones). 

The introduction is the place to make the case that there needs to be integration between frameworks and provide the evidence that this is both possible as well as useful. Then follow with the high level description and definition (above) so that as readers view section 2, they are already familiar with the 5 domains.

The authors also need to clarify how (what approach) they will be using to show how to integrate the SDH with other framework(s), e.g., a review of existing literature and synthesis of ideas about the connections.

Lines 49-50: and these studies are often performed on convenience samples of healthy, middle aged, white humans who don’t represent the full scope of guardians.  Please expand slightly, it will make the argument in the manuscript even stronger. See Saunders Plos ONE 2017 for a nice analysis.

Line 102: note that while this use of the term guardian does fit this use, guardians for children are often temporary.  And the laws and policies we seek to influence consider animals as being owned by the person who looks after them (with variations depending on state and local laws).  Just something for the authors to consider in their choice of terminology.

Line 113: this is likely a dog, as cats and other companions don’t usually need walks.  Please consider the specificity of the terms used to avoid a dog-centric approach.

Line 122: please don’t overinterpret reference 37.  Frame as one example in one location. There are many factors which influence shelter intake including one which doesn’t seem to have been included: proximity to the shelter. And this isn’t a peer reviewed document; also paid for and co-performed by an organization with a vested interest in the results.  Better or additional references might include Patronek AJVR 2010; 71:161-168 and Patronek AJVR 2010; 71: 1321-1330.

Should section 2 be about evidence that SDH can impact animal welfare?

I really like figure 1: I do wonder if there is a way to adjust the “guardian ability” to more directly align with both the SDH and Domain factors.  For example. Domain 1 is nutrition, could “understanding of companion animal needs” be included there?  Then the resources might be “quality nutrition for CA”. Other comments: “Engage in” could include interactions with conspecifics for social CA. And Domain 4, I don’t understand what is meant by “Negative body language and behaviours”. Negative behaviours yes but not the body language. And would destructive behaviors and aggression better fit in domain 5? I’m not sure I’m fully in favor of Domain 5 being only negative external outcomes for the animals.  Seems like it should also include some of the psychological results of poor health and behaviour.

Lines 180-184: I like this disclaimer and would like to see the earlier references in this paragraph either deleted or included with more information about the quality of the study. I don’t want to demonize lower education pet owners as there are many different reasons for the results, highlighted in this paragraph, which may not have anything to do with education (which the authors only present at the end of the paragraph).  I think that a more general way to connect education and animal care/relinquishment would be better here as well as comments on how to impact that.

Lines 208-211: these should be moved to the next section perhaps as a transition or linkage.

Section 2.2: while older, Scarlett et al JAAWS 1999; 2(10): 41-57 specifically focused on relinquishment relative to health and personal issues.  See also Eagan et al Frontiers Vet Sci 2022 for dogs specifically.

Section 2.3: the authors might also consider the book Underdogs: people pets and poverty by Rowan and Arluke 2020.

Section 2.4: first paragraph. This is a mix of what it means and how pets can help. Please create two paragraphs with topic sentences to clarify how this SDH applies and how CA might mitigate. Then transition and link to how CA might make things worse or be impacted negatively by the inability to walk in unsafe locations, for example.

Line 340-1: for some folks with low income, spending inadequate time with their animal isn’t an issue, e.g., fixed income due to disability or retirement. Please edit.

Line 343-345: again, low income isn’t really the “reason”; it is a surrogate for factors in the SDH! Please edit and note that relinquishment continues to be multifactorial.

Line 348: affordability yes, but physical access as well as knowledge about when and how often to get puppies vaccinated are likely other key problems.  Please edit.

Line 394: “Another example” PPHE is already discussed above, please edit this sentence for clarity about what the program does and consider moving the rest of this sentence sooner in the section about PPHE and AlignCare.

Paragraph starting line 407: this is a well constructed example and argument for the linkage. Consider how this sort of example with clear connections and interconnections could be brought to all the SDH sections.

Line 445 just haphazardly reintroduces the idea of other frameworks, presumably one health and one welfare.  But these frameworks were only discussed in the very beginning of the manuscript and not included explicitly elsewhere.  If they are important then they need to be integrated more deeply into sections 2 and 3.  If they are only to set the stage for the in-depth assessment of SDH and 5 Domains then that should be made clear up front and again here instead of this one off mention. This goes to my comment about what is this manuscript really focused on and how is that done that I made at the beginning of this review.  It’s a little unclear.

Figure 2: an additional barrier is taking time off from work and losing income that way. Not sure it is needed but…More importantly, how does figure two, as a figure rather than explanation, link to the argument?  The information in the figure should be what is explained and shared in the sections of text above. Is the figure meant to complement those sections? Be used to help others understand the issues? Illustrate the importance of the manuscript? That isn’t clear in the text.

Lines 460-1:  how does one “address income” and “education levels”? Rather I believe the argument is that understanding these factors’ influences on humans and subsequently their CA is what is important because then interventions can be designed and tested to improve animal welfare (and human welfare) outcomes.

Reviewer 3 Report

Dear Authors, thank you for your valuable work, for your insightful and well-documented manuscript, and for sharing this integrated approach to broaden and improve the perspective of each and every reader. I do not have any comments or correction suggestions, just congratulations.
